# [^18^F]FDG-PET/CT in Idiopathic Inflammatory Myopathies: Retrospective Data from a Belgian Cohort

**DOI:** 10.3390/diagnostics13142316

**Published:** 2023-07-08

**Authors:** Halil Yildiz, Charlotte Lepere, Giulia Zorzi, Olivier Gheysens, Fabien Roodhans, Lucie Pothen

**Affiliations:** 1Department of Internal Medicine and Infectious Diseases, Cliniques Universitaires Saint-Luc, Institute of Clinical and Experimental Research (IREC), Université Catholique de Louvain (UCLouvain), Avenue Hippocrate 10, B-1200 Brussels, Belgium; fabien.roodhans@saintluc.uclouvain.be (F.R.); lucie.pothen@saintluc.uclouvain.be (L.P.); 2Department of Internal Medicine, Hôpital d’Arlon (Vivalia), 6700 Arlon, Belgium; charlotte.lepere@vivalia.be; 3Department of Laboratory, Avenue Hippocrate 10, B-1200 Brussels, Belgium; giulia.zorzi@saintluc.uclouvain.be; 4Department of Nuclear Medicine, Cliniques Universitaires Saint-Luc, Institute of Clinical and Experimental Research (IREC), Université Catholique de Louvain (UCLouvain), Avenue Hippocrate 10, B-1200 Brussels, Belgium; olivier.gheysens@saintluc.uclouvain.be

**Keywords:** idiopathic inflammatory myopathies (IIMs), polymyositis, dermatomyositis, [^18^F]FDG-PET/CT, interstitial lung disease, cancer

## Abstract

[^18^F]FDG-PET/CT is a useful tool for diagnosis and cancer detection in idiopathic inflammatory myopathies (IIMs), especially polymyositis (PM) and dermatomyositis (DM). Data deriving from Europe are lacking. We describe [^18^F]FDG-PET/CT results in a Belgian cohort with IIMs, focusing on patients with PM and DM. All of the cases of IIMs admitted between December 2010 and January 2023 to the Cliniques Universitaires Saint-Luc (Belgium) were retrospectively reviewed. In total, 44 patients were identified with suspected IIMs; among them, 29 were retained for final analysis. The mean age of the retained patients was 48.7 years; 19 patients were female (65.5%). Twenty-two patients had DM and seven had PM. The mean serum creatinine kinase (CK) and the mean CRP levels were 3125 UI/L and 30.3 mg/L, respectively. [^18^F]FDG-PET/CT imaging was performed for 27 patients, detecting interstitial lung diseases (ILDs) in 7 patients (25.9%), cancer in 3 patients (11.1%), and abnormal muscle FDG uptake compatible with myositis in 13 patients (48.1%). All of the patients who were detected to have ILDs via PET/CT imaging were confirmed using a low-dose lung CT scan. Among the patients who were detected to have abnormal muscle FDG uptake via PET/CT scans (13/28), the EMG was positive in 12 patients (*p* = 0.004), while the MRI was positive in 8 patients (*p* = 0.02). We further observed that there was a significantly higher level of CK in the group with abnormal muscle FDG uptake (*p* = 0.008). Our study showed that PET/CT is useful for detecting cancer and ILDs. We showed that the detection of abnormal muscle uptake via PET/CT was in accordance with EMG and MRI results, as well as with the mean CK value, and that the presence of dyspnea was significantly associated with the presence of ILDs detected via PET/CT imaging (*p* = 0.002).

## 1. Introduction

Idiopathic inflammatory myopathies (IIMs) are a group of rare inflammatory disorders that affect skeletal muscle. They can be classified into four subgroups: polymyositis (PM), dermatomyositis (DM), inclusion body myositis (IBM), and immune-mediated necrotizing myositis (IMNM) [1,2]. Autoimmunity plays a key role in IIM pathogenesis, and autoantibodies might be identified in more than 50% of cases [3]. Autoantibodies that are specific to myositis are referred to as myositis-specific antibodies (MSAs), and antibodies that are commonly seen in myositis associated with connective tissue diseases (CTDs) are named myositis-associated antibodies (MAAs) [4]. During the last decade, substantial progress was made in identifying novel autoantibodies. For each MSA, a demonstrated correlation has been shown between specific clinical manifestations, aiding in the diagnosis, classification, and prognosis of patients into more homogeneous groups (phenotyping) [3,4].

IIMs are clinically characterized by progressive proximal muscle weakness and an elevated serum creatine kinase level. Depending on the type of myositis, other symptoms may be associated with IIMs, including fatigue, fever, dysphagia, cough, dyspnea, and arthralgia. Patients with DM also present with cutaneous lesions, including periorbital violaceous rash (also called heliotrope rash), Gottron’s papules, calcinosis, Raynaud’s phenomenon, skin ulcers, mechanic’s hands, or periungual erythema [3,5].

An important clinical presentation of IIMs is antisynthetase syndrome (ASS), which is characterized by the association of Raynaud’s phenomenon, mechanic hands, fever, interstitial lung disease (ILD), and inconstant cutaneous rash. This syndrome was named because of the first described antibody associated with this clinical entity, anti-Jo-1, that targets histidyl tRNA synthetase, which is one of the aminoacyl tRNA synthetases. To date, a total of eight anti-tRNA synthetase autoantibodies (ASAs) have been reported in myositis in addition to anti-Jo-1: anti-PL12, anti-PL7, anti-EJ, anti-OJ, anti-KS, anti-Zo, and anti-Ha [3]. Notably, studies have demonstrated that some of the particular clinical manifestations are associated with each individual ASA [3].

IMNM is also characterized by muscle weakness, but usually without skin or lung involvement. Autoantibodies such as 3-hydroxy-3-methylglutaryl-CoA reductase (HMGCR) or signal recognition particles (SRPs) are found in 66% of patients [6]. Anti-HMGCR autoantibodies have mainly been described in patients who were pre-exposed to a statin [6].

IBM is another form of inflammatory myopathy. It affects patients over 50 years old with a predominance of distal and quadriceps muscles involvement. The progression of this disease is slow, and it does not affect the skin or the lungs. Unfortunately, this subtype of inflammatory myopathy is highly refractory to treatment [7].

PM is more a diagnosis of exclusion, and physicians should always consider IBM, DM, or IMNM before reaching this diagnosis. There are no specific autoantibodies, and muscle biopsy is the only way to definitely confirm the diagnosis [3,8].

Finally, it is important to emphasize that IIMs, particularly PM and DM, may be associated with interstitial lung diseases (ILDs) and cancer [1,2]. As previously noted, myositis autoantibodies are helpful in classifying patients. Antimelanoma-differentiation-associated gene 5 (anti-MDA5) and, as previously indicated, anti-JO-1 autoantibodies and other ASAs are more frequently associated with interstitial lung diseases (ILDs). Antitranscription intermediary factor 1γ (anti-TIF1γ) and antinuclear matrix protein 2 (anti-NXP2) are more frequently associated with cancer.

Electromyography (EMG) and magnetic resonance imaging (MRI) are diagnostic tools that are used to prove muscle involvement; however, those tools are not able to differentiate between myopathies. With EMG, signs of PM and DM are characterized by muscle membrane irritability, including fibrillation potential, positive sharp waves (PSWs), and/or myopathic motor units [9]. The same abnormalities have been described with IBM and IMNM. MRI usually shows intramuscular T2 hypersignaling in DM [10]. However, these findings are also possible with PM and IMNM. MRI is interesting in diagnosing IBM, as the principal IBM characteristics on MRI are the involvement of the thigh and distal muscles [11,12]. Therefore, EMG and MRI are often performed to confirm a suspicion of IIM and, eventually, to select a preferential localization site for a biopsy [13,14]. Indeed, muscle biopsy is the gold standard to confirm the type of IIM, but this technique is invasive and requires an appropriate anatomopathological laboratory. Regarding ILD, high-resolution computed tomography (HRCT) is the gold standard for examination [15].

In the literature, several diagnostic and classification criteria for IIMs have been proposed: Bohan–Peter, Dalakas, ENMC 2004, and EULAR 2017 [2,8,16,17]. The mid-seventies Bohan and Peter’s criteria are still commonly used in clinical practice. Based on a combination of clinical signs (proximal and symmetrical muscle weakness, and/or typical cutaneous changes such as Gottron’s signs), biological abnormalities (increase in serum muscle enzymes), EMG characteristics, and muscle biopsy results, a diagnosis of definite, probable, or possible (dermato)-myositis can be retained. To note, in this list, muscle biopsy is not a mandatory criterium.

In 2004, to the ENMC criteria were added some clinical exclusion criteria (such as ocular weakness of toxic myopathy), including other laboratory criteria such as MRI abnormalities (increased signal on STIR; in other words, edema) and MSA detection, on top of a specific histological pattern in muscle biopsies which became mandatory; this latter criterium makes it increasingly difficult to apply the criteria in clinical practice [17]. More recently, the EULAR/ACR criteria were published in 2017. Combining easily available clinical, biological (MSA and CK enzymes), and histopathological findings in the case of a biopsy, a score is determined. Above a certain cut-off point, depending on the presence of a muscle biopsy, a diagnosis of probable or definite IIM can be retained [8]. It is important to mention that the only MSA/MAA antibody included in the 2017 EULAR/ACR criteria was the anti-JO1 antibody.

The main issue with all these criteria is that they are used either as classification criteria or diagnostic criteria, except for EULAR 2017. Moreover, they do not take into account the recent progress made in autoantibodies detection and in IIM phenotyping regarding autoantibody types.

[^18^F]FDG-PET/CT is a non-invasive imaging technique, using a combination of scanner and radioactive glucose as a tracer, to detect abnormal morphological and functional changes in the whole body. In the context of IIM, especially PM and DM, it became a useful tool for cancer screening [1,18]. Recently, in a systematic review, Bentick et al. showed that PET/CT is performant for the detection of malignancies compared to conventional work-up, with a sensitivity and specificity of 66, 7–94%, and 80–97.8%, respectively [19]. More recently, [^18^F]FDG-PET/CT was also used for IIM diagnosis as well as for the activity assessment of muscle disease and the detection of extra-muscular manifestations, such as ILD [19]. As an example, in the systematic review of Bentick et al., compared to HRCT, the sensitivity of PET/CT for detecting ILD was 93–100% [19]. However, the interpretation criteria for detecting abnormal muscle or abnormal lung uptake vary widely according to various studies. Some studies used a visual assessment, considering abnormal muscle or lung uptake superior to liver or superior to the mediastinal blood pool [20,21,22]. The semi-quantitative and quantitative parameters used by others also differ between what is studied, from SUVmax to mean SUVmax or SUVratios [21,23,24,25,26,27,28,29]. Notably, most data are limited to retrospective studies deriving from China, Japan, Canada, France, or Spain [20,21,22,23,24,25,28,30,31,32]. Finally, despite a recent demonstration of its efficacy, PET/CT is not included in previously detailed diagnoses or classification criteria.

The objective of this study was to retrospectively describe the results of [^18^F]FDG-PET/CT efficiency in a Belgian cohort with IIMs, focusing on patients with PM and DM. The main goals were to evaluate the performance of PET/CT imaging in disease diagnosis (e.g., abnormal muscle uptake) compared to the usual methods, such as MRI and EMG as well as PET/CT performance in ILD and malignancy screening. We also aimed to evaluate an eventual correlation between PET/CT results and clinical or biological abnormalities, such as serum CK levels and dyspnea.

## 2. Materials and Methods

This retrospective study was conducted at the Cliniques Universitaires Saint-Luc in Belgium, and was approved by our local institutional Ethics committee. All cases of IIM admitted between 12/2010 and 01/2023 in the Department of Internal Medicine were retrospectively reviewed. The data were collected using our institutional database (Epic electronic health record) and the database of the Internal Medicine Department. The inclusion criteria were as follows: patients ≥18 years old, dermatomyositis, polymyositis, and overlap syndrome. Overlap syndrome was considered when myositis was associated with either clinical and/or autoantibody overlap feature: scleroderma, sclerodactylia, polyarthritis, or Sjögren, or SSA/SSB, RNP, Ku, or PMScl antibodies. Only patients that fulfilled the Bohan and Peter criteria [2] or EULAR 2017 criteria [8] for the diagnosis of IIM were included in the study. Patients with immune-mediated necrotizing myositis (formerly named statin-induced myositis) and inclusion body myositis were excluded.

We collected the following information: clinical and labs characteristics (including CK, antinuclear antibody, and myositis-specific autoantibodies), electromyography (EMG), whole-body muscle magnetic resonance imaging (MRI), the results of skin and muscle biopsies, 18F-fluorodeoxyglucose positron emission tomography/computed tomography [^18^F]FDG-PET/CT, thoraco-abdominal enhanced CT scan, and treatment and outcome (mortality). All PET/CT scans were performed prior to immunosuppressive therapy, and patients fasted for at least 6 h before [^18^F]FDG injection. The criterium used to interpret abnormal muscle uptake on [^18^F]FDG-PET/CT was the following: the PET/CT was considered positive if FDG uptake was equal or greater than liver uptake [27,33]. The criteria used to interpret muscle biopsies were those from ENMC [17] and EULAR 2017 [8]. The criteria used to interpret EMGs were those well described by Paganoni [9]. EMG was considered positive if it showed signs of muscle membrane irritability: fibrillations potentials, positive sharp waves (PSW), and myopathic motor units [9]. MRI was considered positive if it showed intramuscular T2 hyperintensities [10].

Quantitative variables were reported as mean values, while qualitative values were shown as numbers and percentages. Statistical tests were performed using GraphPadPrism 9 (GraphPad Software, Inc., San Diego, CA, USA). For the comparison of CK values for patients with or without abnormal FDG uptake on the PET/CT, results were reported as the mean and standard deviation. The statistical analysis was performed using a non-parametric test (Mann–Whitney test) after excluding the outliers (ROUT method) and verifying the abnormality of the values distribution. The contingency analysis was performed using Fisher’s exact test. A *p*-value < 5% was considered significant.

## 3. Results

In total, 44 patients were identified to have idiopathic inflammatory myopathies (IIMs); among them, 29 patients were retained for the final analysis (Figure 1).

The mean age was 48.7 years old, and 19 patients were female (65.5%). Twenty-two patients had dermatomyositis and seven had polymyositis. One patient in our cohort had only dermatologic involvement without myopathy. A diagnosis of amyopathic dermatomyositis was retained based on a skin biopsy. The clinical and biological characteristics are presented in Table 1.

The most frequent symptoms were cutaneous lesions (75.8%), muscle weakness (82.7%), dysphagia (31%), cough (17.2%), shortness of breath (17.2%), and arthralgia (17.2%). The antinuclear antibody (ANA) was positive in 21 patients (72.4%), but most patients were of a low titer (≤1/320 in 16 patients (55.1%)). Myositis-specific autoantibodies were found in 58.6% of patients, and are detailed in Table 1.

The mean serum creatinine kinase (CK) and mean CRP levels were 3125 UI/L and 30.3 mg/L, respectively. EMG was performed in 27 patients, and 62.9% showed signs of myositis. Whole-body muscle MRI was performed in 19 patients, and 57.8% showed signs of myositis. [^18^F]FDG-PET/CT imaging was performed in 27 patients, in whom ILD was detected in 7 patients (25.9%), cancer in 3 patients (11.1%), and abnormal muscle FDG uptake compatible with myositis was detected in 13 patients (48.1%). The cancers detected via PET/CT imaging were as follows: one lung cancer, one esophageal cancer, and one ovarian cancer. The cancers that were not detected were one prostate cancer and one large granular leukemia. Three patients with cancer-associated myositis had consistent MSAs: two had PL7 antibodies and one had theTIF1-γ antibody. Among the patients with ILDs observed from PET/CT imaging, all were confirmed via a low-dose lung CT scan (HRCT). Some examples of patients with myositis, ILD, and cancer diagnosed with PET/CT imaging are presented in Figure 2 and Figure 3.

The patients were treated by corticosteroids (93%), methotrexate (62%), azathioprine (31%), mycophenolate mofetil (0.3%), rituximab (0.6%), and IV immunoglobulins (17.2%). Three patients showed complications with opportunistic infections (one endocarditis due to *Candida albicans*, one invasive aspergillosis, and one *Nocardia* infection), and the overall death rate was 13.7%.

We analyzed whether the abnormal muscle uptake disclosed by [^18^F]FDG PET/CT was in accordance with the EMG or whole-body muscle MRI findings. Among the patients with abnormal muscle FDG uptake observed with PET/CT imaging (13/28), the EMG results were positive in 12 patients (*p* = 0.004), while the MRI results were positive in 8 patients (*p* = 0.02) (Table 2).

Among the patients without abnormal FDG uptake (15/28), the EMG results were positive in nine patients while the MRI results were positive in only three patients.

We performed the same analysis with the muscle biopsy results. Among the patients with abnormal muscle FDG uptake observed from PET/CT scans, myositis was confirmed via biopsy in seven patients, while in the other group, myositis was finally disclosed using biopsies in four patients (statistically not significant). We further compared the mean CK values at diagnosis in the two groups of patients, with and without abnormal muscle FDG uptake observed via PET/CT imaging.

Interestingly, we observed a significantly higher level of CK in the group with abnormal muscle FDG uptake (*p* = 0.008) (Figure 4).

Finally, we also wanted to evaluate the correlation between ILD detection by PET/CT imaging and symptoms such as dyspnea. Seven patients in our group were detected to have ILD via PET/CT scans, and all were confirmed with high-resolution chest CT (HRCT). Four of them had symptoms of dyspnea, as none of the twenty PET/CT-negative patients were asymptomatic (*p* = 0.002) (Table 3).

## 4. Discussion

Our study showed that, in our cohort of IIM patients, there was a significant correlation between abnormal muscle signaling found from PET/CT and EMG or MRI (*p* < 0.05), as well as with dyspnea and the presence of ILD in PET/CT imaging (*p* < 0.05). Interestingly, we found that the mean CK values were higher in patients with abnormal muscle uptake in PET/CT scans. Most of existing studies concerning [^18^F]FDG-PET/CT scans in idiopathic inflammatory myopathies (IIMs) were retrospective trials performed in Japan and China [20,21,22,23,24,25,28,30,31,32]; studies deriving from Europe are scarce [26,27,29,34,35], and our study is the first from Belgium.

Several existing studies evaluated FDG uptake in muscle; however, many used inconstant interpretation criteria, such as visual analysis or semi-quantitative analysis (SUVanalysis) [20,21,22,23,24,25,26,27,28,29,30,31,32]. The studies of Owada, Tateyama, and Motegi et al. used visual analyses to assess FDG uptake in muscle [20,21,22]. In the report of Owada, FDG uptake correlated with EMG abnormalities [20]. In the study of Motegi et al., as in our study, FDG uptake correlated with MRI results and CK levels [22]. On the other hand, Tateyama et al. showed no correlation between FDG uptake in muscle and MRI results or CK levels [21]. Nevertheless, in these three reports, the visual criteria used were not similar. In the study of Owada, muscle uptake was considered positive if uptake was equal to or higher than the liver uptake, while in the two others, uptake was considered positive if equal or greater than the mediastinal blood pool. Concerning studies using semi-quantitative analyses (SUV max, mean SUV max, and SUV ratios), the results are also inconstant, with some reports showing a muscle FDG correlation to the CK level and/or the MRI findings [23,24,25,26] and others showing no correlation [21,27,28,29]. It is important to highlight that in most of these studies, patients were already on corticosteroids before undergoing PET/CT scans, which can interfere with FDG uptake; this was not the case in our cohort of patients. There is also a lack of standardization in the interpretation criteria for using semi-quantitative methods in these previous studies. Notably, only one study compared visual analysis to semi-quantitative methods and showed similar diagnostic accuracy [24].

Altogether, this demonstrates that there is a crucial need for a prospective controlled trial with standardized interpretation criteria and optimal patient preparation. Recently, a promising diagnostic modality, [^18^F]FDG-PET/MRI, was developed. This technique has the advantage of combining PET to magnetic resonance imaging [36]. Its sensitivity ranges from 60 to 100% and its specificity from 50 to 100%, and is highly dependent on the parameters used to quantify FDG uptake. The usefulness of this interesting technique in IIM also needs to be confirmed by a prospective trial.

In our cohort, three patients were diagnosed with cancer-associated myositis via PET/CT (one esophageal cancer, one lung cancer, one ovarian cancer). The value of [^18^F]FDG-PET/CT for cancer screening has been demonstrated in several studies with a high negative predictive value [30,34,35]. Callaghan et al. also showed that PET/CT has the same ability to detect cancer compared to conventional work-up (thoraco-abdominal enhanced CT scan, colonoscopy, gastroscopy, etc.) [34]. However, false negative results are also possible, particularly in situations of poorly avid lesions, small-sized tumors (e.g., below camera’s spatial resolution) [37], traditionally PET/CT-negative cancers (prostate or renal cancer), or non-solid cancers, as in one of our patients.

Seven patients within our group were detected to have ILD via PET/CT, and all were confirmed using high-resolution chest CT (HRCT). Interestingly, dyspnea was significantly associated with the presence of ILD via PET/CT, as was demonstrated in Owada et al.’s study [20]. Several studies have demonstrated the ability of PET/CT to detect ILDs, and showed a good correlation with HRCT [21,22,23,28,31,32]. The antimelanoma-differentiation-associated gene 5 (MDA5) antibody is a well-known IIM antibody associated with rapidly progressive ILD (RP-ILD) [38]. Identifying RP-ILD is highly important, since its mortality rates range from 19 to 27%, with the 3-month mortality rate being over 30% [39,40,41]. Three previous studies used semi-quantitative methods to identify patients that may develop this potentially lethal complication [23,31,42]. Li Y and Liang J et al. both showed that the mean lung SUV can predict RP-ILD development. Unfortunately, the lung SUV cut-off value was not similar in these two studies [23,31]. Recently, Zhang et al. showed that the PET score may be more useful than SUV for evaluating pulmonary disease activity [42]. The PET score is a five-point visual scale determined in six lung zones based on the maximum FDG uptake: one (no uptake), two (uptake ≤ mediastinum uptake), three (uptake > mediastinum but <liver uptake), four (uptake moderately above liver uptake), and five (uptake markedly above liver uptake). However, in this study, the majority of patients were already undergoing immunosuppressive therapy while subject to PET/CT, leading to a potential bias in the results of lung FDG uptake. To reiterate, this highlights the need for a larger prospective cohort.

Our study has several limitations. Firstly, as there are few studies on IIM [20,24,25] due to the rareness of this disease, we had to perform a retrospective analysis on a small number of patients, and we did not include control patients. Nevertheless, our results align with a few other European cohorts of PET/CT in IIM [26,34,35].

Secondly, we did not perform a thorough comparison of muscle FDG uptake detected by PET/CT and edema or by abnormal MRI signals. This localization concordance analysis was not possible for several reasons, such as delays between the imaging technique for some patients and differences in the machine used. Detailed analyses comparing the two imaging modalities would be interesting to conduct, in order to confirm the good correlation we observed.

Finally, we were not able to show a significant correlation between PET/CT and muscle biopsy, which remains the “gold standard” for the diagnosis of autoimmune myositis. This was possibly due to the small number of biopsies performed in our cohort (*n* = 15). As a reminder, we used the Bohan and EULAR 2017 criteria to decide on which patients to include in our study; for both criteria, a muscle biopsy is not mandatory for IIM diagnosis [2,8].

## 5. Conclusions

Our study showed that PET/CT is useful for cancer and ILD detection, but we also showed that abnormal muscle uptake on PET/CT was significantly in accordance with EMG and MRI. Notably, this is the first study of this kind from Belgium. Moreover, we showed that the mean CK value was higher in patients with abnormal uptake in PET/CT imaging. We also showed that the presence of dyspnea was significantly associated with the presence of ILD via PET/CT scans. Based on our results, a prospective trial could be designed in future to confirm the correlation we observed between PET/CT results and those of other diagnosis tools (EMG, MRI, and CK values).

## Figures and Tables

**Figure 1 diagnostics-13-02316-f001:**
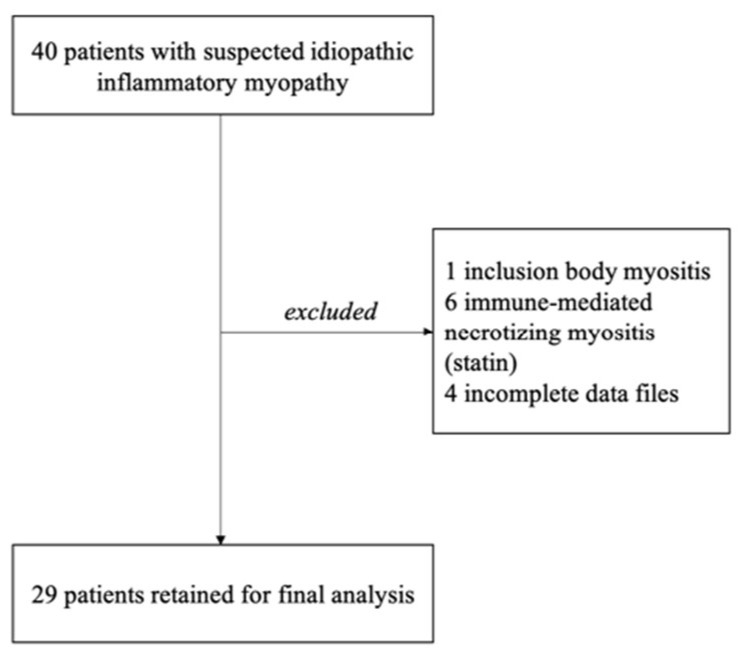
Study flowchart.

**Figure 2 diagnostics-13-02316-f002:**
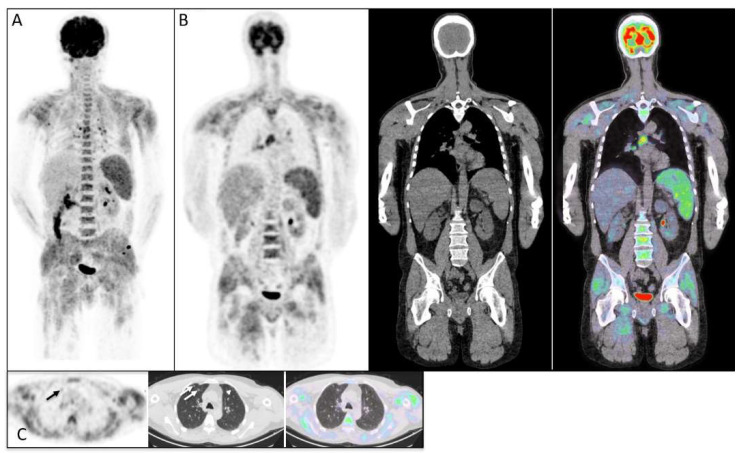
Maximum intensity projection (MIP) PET image (**A**) and coronal FDG-PET, CT, and fused PET/CT images (**B**) showing a heterogeneously increased muscular FDG uptake, multiple hypermetabolic mediastinal lymphadenopathies, and splenic hypermetabolism in a patient with dermatopolymyositis. (**C**) Transaxial FDG-PET, CT, and fused PET/CT images showing slightly hypermetabolic pulmonary nodular condensations (arrow) and non-hypermetabolic subpleural reticulations (arrowhead). Furthermore, heterogeneously increased muscle uptake is present in the context of dermatomyositis.

**Figure 3 diagnostics-13-02316-f003:**
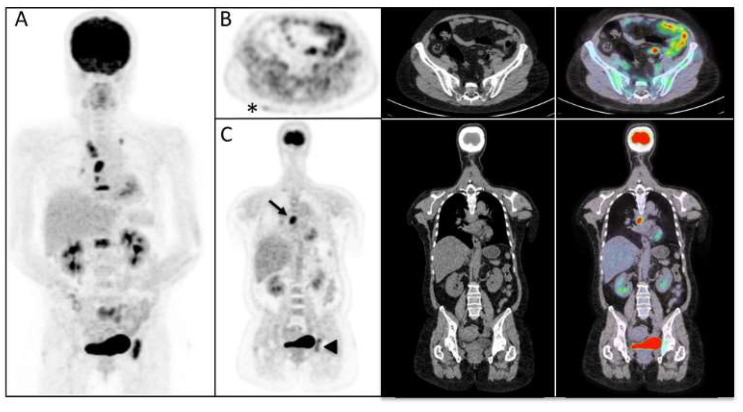
Maximum intensity projection (MIP) PET image (**A**), transaxial (**B**), and coronal (**C**) FDG-PET, CT, and fused PET/CT images demonstrating multiple hypermetabolic mediastinal lymphadenopathies (arrow) and bone lesions (arrowhead) in a patient with lung cancer. Furthermore, a slightly increased cutaneous FDG uptake was observed at the right lumbar region (asterisk). Cutaneous biopsies confirmed the diagnosis of dermatomyositis.

**Figure 4 diagnostics-13-02316-f004:**
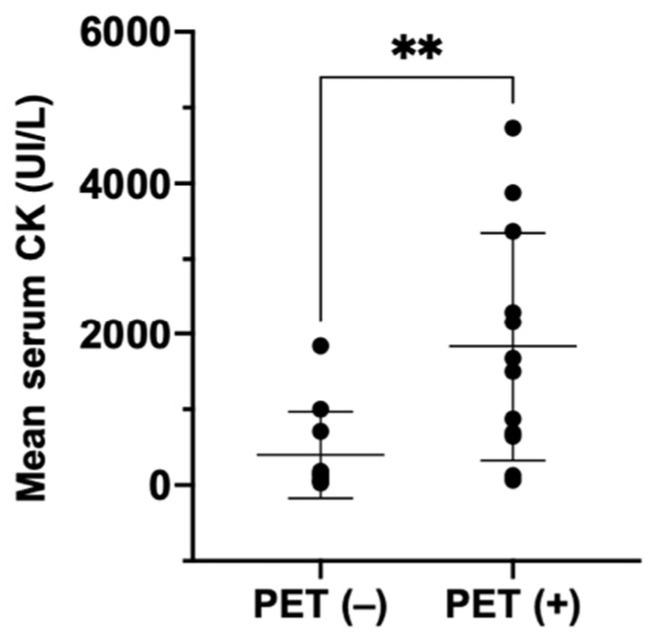
Comparison of mean CK values (UI/L) at diagnosis between PET-negative (no muscle uptake on PET/CT) and PET-positive (abnormal muscle uptake on PET/CT) IIM patients. ** *p* = 0.008.

**Table 1 diagnostics-13-02316-t001:** Clinical characteristics of patients with idiopathic inflammatory myopathies (IIMs).

Total number of patients	29
Mean age at diagnosis (years)	48.7
Sex (female)	19 (65.5%)
Myositis subset	
Dermatomyositis	22 (75.8%)
Polymyositis	7 (24.1%)
Overlap syndrome	0 (0.0%)
Symptoms	
Muscle weakness	24 (82.7%)
Skin lesions	22 (75.8%)
Cough	5 (17.2%)
Dyspnea	5 (17.2%)
Dysphagia	9 (31%)
Arthralgia	5 (17.2%)
Positive ANA	21 (72.4%)
ANA, titer	
1/80	8 (27.5%)
1/160	4 (13.7%)
1/320	4 (13.7%)
1/640	2 (0.6%)
1/1280	3 (10.3%)
Myositis-specific autoantibodies (17/29)	
SAE	2 (0.6%)
SRP	0
MDA5	1 (0.3%)
MI-2	1 (0.3%)
Ro52	0
NXP2	2 (0.6%)
TIF1-γ	4 (13.7%)
PI12	1 (0.3%)
PI7	3 (10.3%)
JO-1	3 (10.3%)
CK level (U/L) at diagnosis (mean) [range]	3125.14 [27–18,760]
CRP level (mg/L) at diagnosis (mean)	30.3
Positive electromyography	17/27 (62.9%)
Positive whole-body muscle MRI	10/19 (52.6%)
Muscle activity on [^18^F]FDG-PET/CT	13/27 (48.1%)
Positive skin biopsy	11/17 (64.7%)
Positive muscle biopsy	11/15 (73.3%)
Interstitial lung diseases	8 (27.5%)
Cancer detected by [^18^F]FDG-PET/CT	3/27 (11.1%)
Treatment	
Corticosteroids	27 (93%)
Methotrexate	18 (62%)
Azathioprine	9 (31%)
Mycophenolate mofetil	1 (0.3%)
Rituximab	2 (0.6%)
IVIG	5 (17.2%)
Opportunistic infection	3 (10.3%)
Death	4 (13.7%)

ANA: antinuclear antibody; CK: creatine kinase; MRI: magnetic resonance imaging; IVIG: intravenous immune globulin (IVIG).

**Table 2 diagnostics-13-02316-t002:** Comparison of [^18^F]FDG-PET/CT muscle uptake findings with electromyography (EMG) and whole-body muscle magnetic resonance imaging (MRI).

	PET/CT-Positive	PET/CT-Negative	*p* Value
**EMG (+)**	12	4	0.0036
**EMG (−)**	1	9	
**MRI (+)**	8	3	0.0198
**MRI (−)**	1	7	

PET/CT-positive: uptake = or > than liver uptake.

**Table 3 diagnostics-13-02316-t003:** Comparison of [^18^F]FDG-PET/CT lung uptake findings with clinical signs of IDL (dyspnea).

	PET/CT-Positive	PET/CT-Negative	*p* Value
**Dyspnea (+)**	4	0	0.002
**Dyspnea (−)**	3	20	

PET/CT-positive: lung uptake = or > liver uptake.

## Data Availability

All of the data presented in this study are available on request from the corresponding author. The data are not publicly available due to privacy restrictions.

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
