# Peer review of "[18F]FDG-PET/CT in Idiopathic Inflammatory Myopathies: Retrospective Data from a Belgian Cohort"

_diagnostics, 2023, doi:10.3390/diagnostics13142316_

Round 1
Reviewer 1 Report
Comments to Author:
The authors revealed the utility of FDG-PET/CT on idiopathic inflammatory myopathy (IIM). The findings of this paper would contribute to future development of diagnosis of IIM
, however, I would like the authors to clarify some points.
1. Material and Methods section: the definition of ‘overlap syndrome’ could not be found.
2. It is well known that patients with dermatomyositis (DM) often complicate cancer, especially DM positive for anti-TIF1γantibody. Autoantibody profiles of the three patients who complicated cancer should be shown.
3. It can be seen that the mean value of CK was elevated in Table 1. Referring to Figure 4, however, it seems that patients with not so elevated CK levels (within normal range?)are included. The range of CK should be shown in Table 1. If patients with clinically amyopathic dermatomyositis, it should be denoted in Materials and Methods and Results sections.
Author Response
Reviewer 1:
The authors revealed the utility of FDG-PET/CT on idiopathic inflammatory myopathy (IIM). The findings of this paper would contribute to future development of diagnosis of IIM, however, I would like the authors to clarify some points.
- Material and Methods section: the definition of ‘overlap syndrome’ could not be found.
=> we add the definition in Material and Methods (line 147): “Overlap syndrome was considered when myositis was associated with ether clinical and/or autoantibody overlap feature: scleroderma, sclerodactylia, polyarthritis, Sjögren, … or SSA/SSB, RNP, Ku or PMScl antibodies”. To note, no patient in our cohort fulfilled this definition (see table 1)
- It is well known that patients with dermatomyositis (DM) often complicate cancer, especially DM positive for anti-TIF1γantibody. Autoantibody profiles of the three patients who complicated cancer should be shown.
=> we add this information in the main text (line 200): “3 patients with cancer associated myositis had consistent MSAs: 2 had PL7 antibodies, and one had TIF-? antibody.”
- It can be seen that the mean value of CK was elevated in Table 1. Referring to Figure 4, however, it seems that patients with not so elevated CK levels (within normal range?)are included. The range of CK should be shown in Table 1. If patients with clinically amyopathic dermatomyositis, it should be denoted in Materials and Methods and Results sections.
=> we add the range in the table 1, and precise in Result that we included one patient with amyopathic dermatomyositis, see line 181: “One patient in our cohort had only dermatologic involvement without myopathy, amyopathic dermatomyositis diagnosis was retained based on skin biopsy”. It is important to note that some patients with clinical myositis (with muscle weakness and pain) have no biological rhabdomyolysis (eg. normal CK level) at diagnosis but pathological EMG, Pet CT, MRI, and/or MRI. We add also a comment in the statistical part of the section regarding CK analysis (line 170 “after outlier exclusion by ROUT method”).

Reviewer 2 Report
This is an interesting study that highlights the potential role of PET/CT in the diagnosis of idiopathic inflammatory myositis.
The authors showed a good correlation between PET/CT and electromyogram, magnetic resonance imaging, and the presence of interstitial lung disease. Abnormal uptake on PET/CT was associated with higher CPK values.
I have no major points of concern about this elegant study. It is mainly descriptive but the results are very informative. On the other hand, the authors clearly discussed potential limitations.
I have a single question regarding the potential value of PET/CT in differentiating polymyositis/dermatomyositis (including antisynthetase syndrome and MDA5)/immune-mediated necrotizing myositis from inclusion body myositis. Could you clearly identify increased distal uptake in patients with inclusion body myositis? The answer to this question by no means reduces the relevance of the study.
Author Response
Reviewer 2
This is an interesting study that highlights the potential role of PET/CT in the diagnosis of idiopathic inflammatory myositis.The authors showed a good correlation between PET/CT and electromyogram, magnetic resonance imaging, and the presence of interstitial lung disease. Abnormal uptake on PET/CT was associated with higher CPK values.
I have no major points of concern about this elegant study. It is mainly descriptive but the results are very informative. On the other hand, the authors clearly discussed potential limitations.I have a single question regarding the potential value of PET/CT in differentiating polymyositis/dermatomyositis (including antisynthetase syndrome and MDA5)/immune-mediated necrotizing myositis from inclusion body myositis. Could you clearly identify increased distal uptake in patients with inclusion body myositis? The answer to this question by no means reduces the relevance of the study.
- we thank the reviewer 2 for its comments. As we notified in our Material et Method section and Flowchart, we didn’t include patient with inclusion body myositis (IBM) in our analysis. Therefore, we are not able to answer to this question. In literature, it is not clear if FDG-PET/CT can be helpful in patients with IBM. However, as we highlighted in our discussion, description and comparison of the localization of FDG uptake could be an interesting topic for further retrospective or prospective studies (including patients with IBM).

Round 2
Reviewer 1 Report
(There are no comments)